# A multivalent binding model infers antibody Fc species from systems serology

**Armaan A. Abraham**[1], **Zhixin Cyrillus Tan**[2], **Priyanka Shrestha**[3], **Emily R. Bozich**[1], **Aaron S. Meyer**[1,2]*

**1** Department of Bioengineering, University of California, Los Angeles, California, United States of America, **2** Bioinformatics Interdepartmental Program, University of California, Los Angeles, California, United States of America, **3** Stanford University, Stanford, California, United States of America

* ameyer@ucla.edu

**Data Availability Statement:** There are no primary data in this paper; all code is available at https://github.com/meyer-lab/mechanismSerology and we

## Abstract

Systems serology aims to broadly profile the antigen binding, Fc biophysical features, immune receptor engagement, and effector functions of antibodies. This experimental approach excels at identifying antibody functional features that are relevant to a particular disease. However, a crucial limitation of this approach is its incomplete description of what structural features of the antibodies are responsible for the observed immune receptor engagement and effector functions. Knowing these antibody features is important for both understanding how effector responses are naturally controlled through antibody Fc structure and designing antibody therapies with specific effector profiles. Here, we address this limitation by modeling the molecular interactions occurring in these assays and using this model to infer quantities of specific antibody Fc species among the antibodies being profiled. We used several validation strategies to show that the model accurately infers antibody properties and then applied the model to infer previously unavailable antibody fucosylation information from existing systems serology data. Using this capability, we find that COVID-19 vaccine efficacy is associated with the induction of afucosylated spike protein-targeting IgG. Our results also question an existing assumption that controllers of HIV exhibit gp120-targeting IgG that are less fucosylated than those of progressors. Additionally, we confirm that afucosylated IgG is associated with membrane-associated antigens for COVID-19 and HIV, and present new evidence indicating that this relationship is specific to the host cell membrane. Finally, we use the model to identify redundant assay measurements and subsets of information-rich measurements from which Fc properties can be inferred. In total, our modeling approach provides a quantitative framework for the reasoning typically applied in these studies, improving the ability to draw mechanistic conclusions from these data.

## Author summary

Antibodies play an important role in our immune response by binding to pathogens and engaging other immune cells to eliminate threats. Mounting evidence points toward the importance of which immune cells are being engaged by antibodies in determining the

have archived it on Zenodo (DOI: 10.5281/zenodo.13942366).

**Funding:** This work was supported by NIH U01-AI148119 to A.S.M. The funders had no role in study design, data collection and analysis, decision to publish, or preparation of the manuscript.

effectiveness of an immune response. While sophisticated experimental methods such as systems serology have been developed to broadly profile the engagement activities of antibodies, determining the presence of antibody structural features relevant to immune engagement remains challenging. Our study addresses this gap by developing a computational model that interprets data from systems serology, allowing us to infer detailed, engagement-relevant structural information about antibodies that are difficult to measure directly. We applied our approach to existing data from COVID-19 and HIV studies, revealing new insights into how antibody structure relates to vaccine efficacy and disease progression. For instance, we found that COVID-19 vaccine effectiveness is linked to the production of certain antibodies lacking a sugar residue called fucose. Our model also helps identify which measurements in systems serology are most informative, potentially streamlining future studies. This work enhances our ability to understand antibody function in disease and may guide the development of more effective antibody-based therapies.

## Introduction

Antibody-mediated protection is central to immunity and important in autoimmunity, infection, therapeutic vaccination, and administered antibody therapies. Therapies are often optimized primarily considering the binding to foreign antigens via the fragment antigen-binding (Fab) region of the antibody, as measured through antibody titer, Fab-antigen binding affinity, and neutralization capability. However, antibody-mediated protection also arises through secondary interactions with immune cells via the fragment-crystallizable (Fc) region of the antibody [1]. While more challenging to quantify and identify as the mechanism of protection, these downstream immune system responses (i.e., effector functions), such as antibody-dependent cellular cytotoxicity [2,3], complement deposition [4], and cellular phagocytosis [5] often hold equal or greater importance than antibody neutralization.

Systems serology jointly profiles the antigen-binding and Fc properties of antibodies [6]. In such assays, antibodies are first captured based on their binding to bead-coupled antigen, leaving antibodies that are specific to that antigen on the bead [7,8]. Next, the binding of those antigen-specific antibodies to a panel of immune receptors—commonly Fc receptors (FcRs)—is quantified. Other molecular properties of the antigen-specific antibody population that influence effector response induction, such as Fc glycosylation, may be quantified in parallel [7–9]. Overall, the binding of several types of biomolecules—such as FcRs and subclass-specific detection antibodies—to this antibody fraction may be measured, here referred to as "detections". By accounting for both necessary events for effector response—antigen and immune receptor engagement—these measurements have proven to be highly predictive of effector cell-elicited responses and antibody-mediated immune protection [10–12].

Systems serology has excelled at identifying Fc feature correlates relevant to disease response. Unsupervised machine learning (ML) approaches, such as PCA, UMAP, network analysis, and tensor decomposition, have found coordinated changes in groups of detections that define subsets of patients with distinct disease outcomes [10,11,13,14]. Supervised ML approaches, such as logistic regression and partial least squares regression (PLSR) have found groups of detections that separate subjects by disease status [10,11,14]. There have also been a very limited number of mechanistic modeling approaches to this data [15]. Their main advantage over ML approaches is their ability to propose quantitative and concrete approaches to

achieving counterfactual effector response profiles (e.g., to increase FcγRIIIa binding by X, one could increase the IgG1 concentration by Y) [15]. However, while the extrapolation capabilities made possible by such mechanistic approaches have been valuable, these approaches have not attempted to infer structural Fc features (such as glycosylation) that aren't already explicitly available in systems serology data. Overall, structural and mechanistic inferences of antibody Fc features driving immune effector responses has largely been limited to manual interpretation of these model outputs.

Mechanistically defining the molecular features that drive outcomes in disease, or therapeutic effectiveness, would enable engineered and personalized interventions. There are a few structural characteristics of the antibody Fc domain that have substantial effects on the binding affinity of the antibody to immune receptors, and thus are particularly influential to effector response induction. These include antibody type (IgG, IgM, IgA, IgD, and IgE), subclass (IgG1, IgG2, IgG3, IgG4), and glycosylation [16,17]. While isotype- and subclass-specific detections provide a direct measure of certain molecular features, antibody glycosylation is usually only indirectly assessed through interaction with Fc receptors [10,11,14,18]. Additionally, while certain Fc species may be quantified in this matter, immune receptor binding is dependent on convoluted and nonlinear changes of several Fc species, which limits the explanatory capacity of these species-specific measurements. Previous mechanistic modeling studies have helped identify antibody Fc features relevant to disease outcomes, but they have ignored the dramatic contribution of glycosylation and multivalent binding to these measurements, which will be essential for accurate inferences [15].

Here, we address this limitation by mechanistically modeling the multivalent binding interactions that occur in systems serology measurements and then using this model to quantify specific antibody Fc species. The model can infer the abundance of antibody Fc features that, at present, lack a straightforward means of quantification by systems serology. For example, the binding model can infer Fc glycosylation with higher throughput, greater antigen breadth, and weaker instrumentation requirements than other measurement techniques such as capillary electrophoresis and LC-MS. We apply these capabilities to derive new observations regarding the properties of SARS-CoV-2 and HIV infection. Additionally, the model allows for the evaluation of the information content of types of detections commonly used in systems serology and, consequently, ways to optimize these assays.

## Results

### A multivalent binding model for quantifying antibody Fc species in systems serology

In systems serology assays, an immune complex consisting of a bead, antigen, and antibodies is incubated with a panel of fluorescently tagged Fc receptors, subclass-specific antibodies, or other binding reagents ("detections") which each bind in a manner dependent on the bound antibody composition. Consequently, the amount of signal observed for a given sample across all the detection wells is a function of the properties and abundance of each antibody Fc species immobilized by the antigen. Our model aims to observe these detection signals as inputs and output the quantity of each antibody Fc species in the antigen-bound immune complex (Fig 1A). The model provides a general interface whereby an arbitrary set of detections and Fc species is provided, along with their binding affinities to one another, which allows a mode of exploratory analysis similar to systems serology itself.

Interpretation of systems serology experiments is complicated by the importance of multivalent binding; valency is critical to both antibody function generally and enabling operation of the assay. Low-affinity FcRs only bind in a multivalent context, where there are

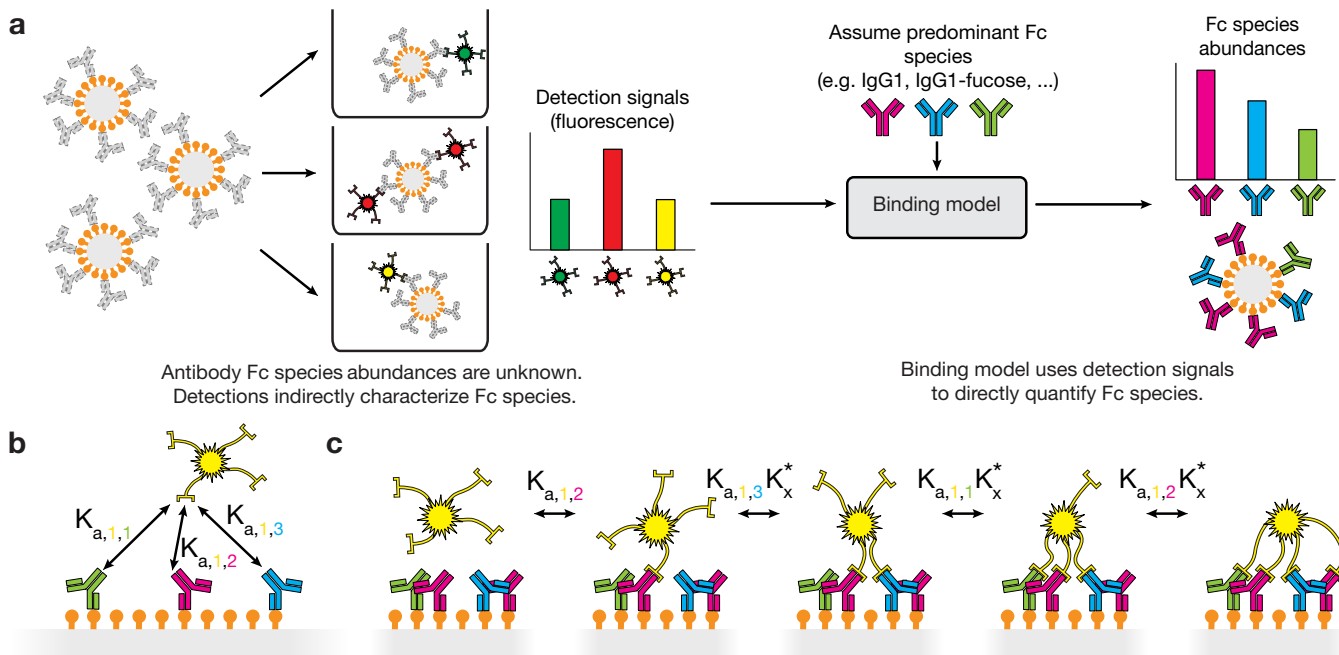

**Fig 1. Binding model usage and implementation. (a)** In systems serology, antibodies are first captured with antigen-coated beads, resulting in the complexes shown on the very left. These complexes are then separated into wells and incubated with different fluorescently tagged detection reagents, each of which leads to a certain amount of bead-associated fluorescence proportional to the amount of binding. The binding model takes these detection signals and infers the abundance of each antibody Fc species immobilized in the complexes. **(b)** Each detection has a known or fit binding affinity to each antibody Fc species, which can be directly used to quantify the equilibrium constant for the initial monovalent binding event. **(c)** To model multivalent binding, we consider all binding events which lead to a particular binding configuration. The monovalent binding event is quantified with the monovalent binding affinity, and subsequent binding events for the same detection are quantified using the monovalent binding affinity multiplied by a crosslinking constant, $K_x^*$, which encapsulates steric and local concentration effects.

multiple antigen-bound, FcR-binding antibodies on an immune complex. Thus, to enable the detection of FcR interactions, they are tetramerized using streptavidin [7,8]. One must therefore model the binding of each FcR to antibodies based on their known affinity to each Fc species, and their capacity to bind multivalently, for accurate inferences (Fig 1B). To quantify arbitrary binding configurations, the model uses the equilibrium constant for each binding event leading up to that configuration (Fig 1C). To quantify steric effects, we introduce a crosslinking constant, $K_x^*$, which scales the known monovalent binding affinity to produce the equilibrium constant for binding events involving an already-bound detection. To tie these quantities of binding configurations (microstates) to the overall detection signal (macrostate) we see, we sum over binding configurations. We have previously applied this formulation extensively to model immune complex binding and subsequent effector responses [19,20].

There are a few additional parameters, known with varying degrees of certainty, that help describe a particular experiment. First, the model requires the total concentration of each detection, which is provided in each of the datasets we analyzed here. As mentioned, the model also requires the crosslinking constant, $K_x^*$. Because $K_x^*$ has not been explicitly measured for streptavidin-based multivalent complexes, we estimated this quantity based on a previous estimate of $K_x^*$ for multivalent immune complexes consisting of antibody-coated antigen. Additionally, the valency of each detection is required, and this information is available in each study (typically 4 for FcR detections and 2 for subclass detections).

## Binding model accurately infers antibody Fc species abundances and is robust to noise

Given that our goal is to infer the abundance of specific Fc species, in part because they are difficult to quantify directly, a straightforward approach to validating these quantities is challenging. Therefore, we instead devised several approaches to indirectly corroborate model inferences.

We first verified that, with synthetic data, we could exactly infer randomly generated starting antibody abundances. Briefly, random abundances were generated for each Fc species by sampling from a log-normal distribution based on that of the subclass detection signals (S1 Fig and S1 Text), and these were used as inputs to the antibody-to-signal function to infer the detection signals. These detection signals were then used as inputs to the signal-to-antibody function to infer the antibody abundances, which were compared to the starting quantities (Fig 2A). This is a useful test because the antibody-to-signal function has been validated and utilized in previous work [19,20]; verifying that the signal-to-antibody function is its inverse supports its accuracy. We found exact agreement between these values ($R^2 = 1.0$), supporting that our implementation could reconstruct these quantities (Fig 2B).

We next tested to what extent noise disrupted accurate inference. With the same method described above, we added noise to the generated detection signals before antibody inference (Methods). This noise emulates error in the detection measurements during systems serology itself. Our model is not designed as a noise-removal tool, and thus we should not test it as such. Instead, our question here is whether the model performs acceptably in the presence of noise (e.g., by verifying that the model does not amplify the noise). We found that our fitting strategy continues to generate accurate inferences even in the presence of noise (Fig 2C and 2D).

The binding model uses the binding affinities between each Fc species and FcR to generate inferences. These quantities have been measured in previous work (Table 1) [21–23]. There is uncertainty in these measurements owing to experimental error and the difficulty of preparing pure solutions of individual species, reflected by their variation across studies. Consequently, we sought to evaluate how this uncertainty impacts the performance of our model. Using the same method, we instead applied noise to the binding affinities only when computing the antibody-to-signal function. We observed accurate inferences even with this error in the affinities (Fig 2E). The run-to-run variation was much larger when adding noise to certain binding affinities as compared to the synthetic detection signals, indicating that certain binding affinities are more sensitive to perturbation. Indeed, when evaluating the sensitivity of the accuracy of the inferences for each Fc species to variation in the binding affinities, we see a sparse relationship (Fig 2F). In other words, most binding affinities have little effect, while a few binding affinities, such as that between IgG3 and FcγRIIIa, have a relatively large effect.

As previously mentioned, our model requires a crosslinking constant parameter, $K_x^*$, which we estimated based on previous experimental data. We used our synthetic validation setup to determine how much the inaccuracy of our $K_x^*$ estimate might affect the accuracy of our model (S2 Fig). We found the model to be robust when provided $K_x^*$ estimates which differed from the ground truth by up to 2 orders of magnitude in either direction, which convinced us that our approach to estimating $K_x^*$ was likely acceptable.

To validate our inferences further, we performed a cross-validation experiment to test the model's ability to infer masked detection signals in real systems serology data. We first chose a detection (such as α-hIgG1 IgG or FcγRIIa) and then masked the signal for that detection in a randomly selected subset of sample-antigen pairs (Fig 3A). We then used this incomplete data to infer the antibody abundances, and in turn predict the masked signals (Fig 3B). In other

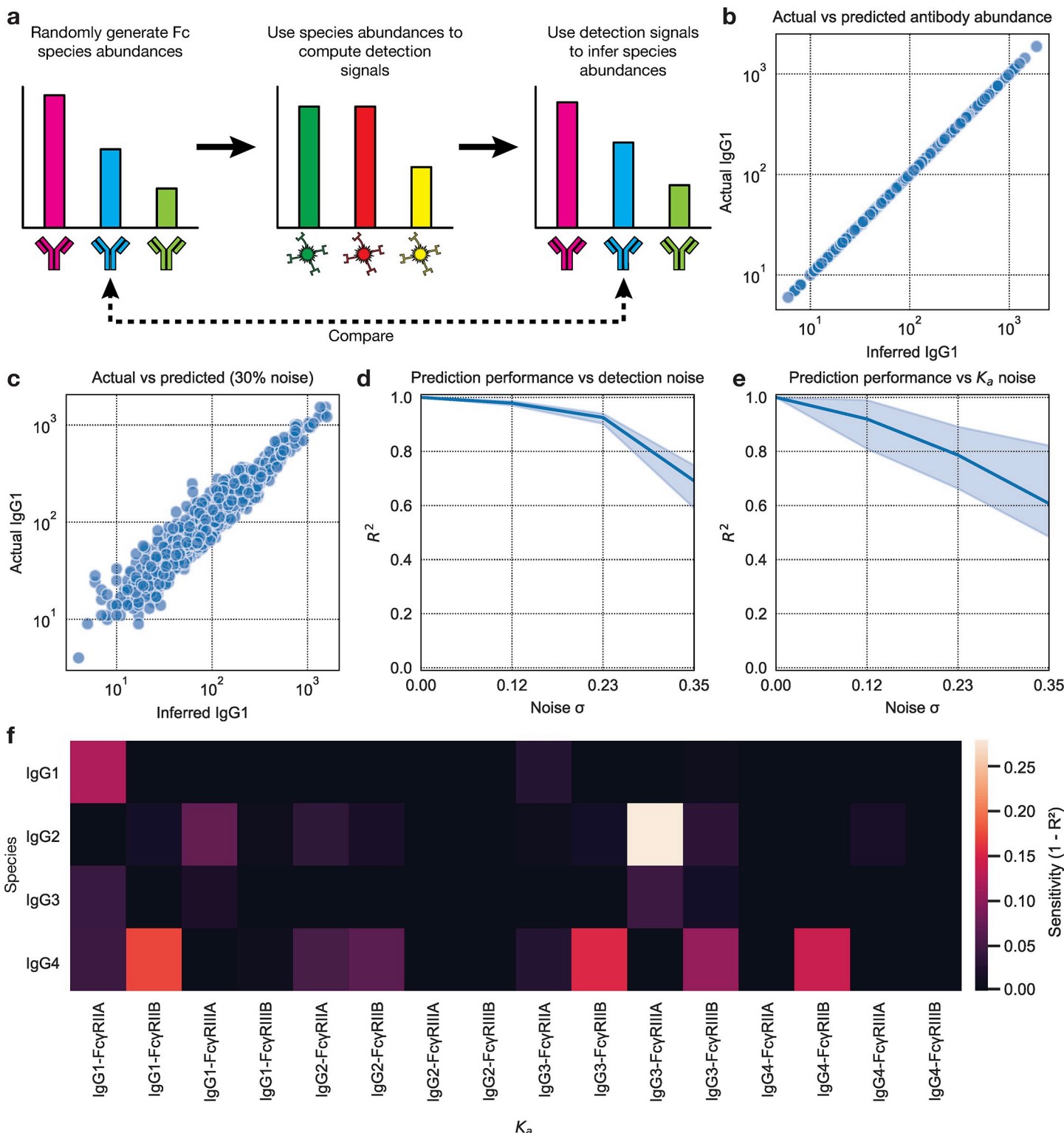

**Fig 2. The binding model accurately infers antibody Fc abundances from synthetic data.** (a) Each Fc species was assigned a random abundance. These abundances were used to generate synthetic detection signals which were then used to infer the original antibody abundances. (b), (c) Initial versus inferred antibody abundance (b) without and (c) with added detection signal noise. (d), (e) The coefficient of determination between the initial and inferred antibody abundances versus the amount of noise added to the (d) detection signals or (e) binding affinities. (f) Each binding affinity was individually perturbed by 30% up and down. The set of affinities with this perturbed affinity was used when computing synthetic signals for randomly generated initial antibody abundances, and the set with the unperturbed affinity for inferring

antibody abundances from the synthetic signals. The sensitivity of the inferences to perturbations of this binding affinity was computed as one minus the agreement ($R^2$) between the inferred and initial antibody abundances, shown separately for each species.

words, the model infers the quantities of the antibodies in the complex based on a subset of the detections, which it then uses to simulate how much of the left-out detection would bind to the complex. We assume that this capability reflects accurate inferences of the underlying antibody abundances.

We ran these imputation tests on systems serology data from a SARS-CoV-2 study by Zohar et al. [11] (Fig 3C). The study involved profiling the antibodies of 193 patients hospitalized with COVID-19, ranging from moderate to severe cases. To contextualize the imputation performance of the binding model, we compared it to that of principal components analysis (PCA). PCA uses the assumption that the input data is a combination of one or more linear components, or patterns, to impute missing values. We masked 10% of the measurements corresponding to a single detection and evaluated each method's ability to impute these values (Fig 3C, 3D and 3E). We first determined the optimal PCA rank for imputation, which was 1 (S3 Fig), and used this optimal rank to compare to the binding model. We saw that the binding model imputes well when measured by the Pearson correlation. However, the accuracy measured by the coefficient of determination was lower. Together, this indicates that the binding model's inferences are linearly related to the ground truth, but that there is a consistent bias. Indeed, this is apparent when plotting the inferences directly (Fig 3C). Interestingly, the α-hIgG1 IgG signal is poorly imputed by both methods, indicating that this information is difficult to infer from the other detection signals. It should be reemphasized that these tests are primarily to indirectly verify the underlying antibody inferences of the binding model, for which an imputation performance roughly on par with PCA is sufficient.

The binding model can additionally impute values for an entirely unobserved detection (Fig 3F), so long as the binding affinities of that detection to each Fc species are known. This capability is not shared by PCA. We tested this capability by incrementally masking values for a single detection and evaluating imputation accuracy at each step (Fig 3G and 3H). We also measured how the imputation performance depends on dataset size at a fixed fraction of missing values and found that the performance is largely invariant to total dataset size (S4 Fig). We see that the model accurately imputes each detection even when 100% of the measurements are masked. However, both FcγRIIa and FcγRIIb gain a significant bias in their predictions as the number of masked values is increased, as indicated by the increasing $R^2$. This likely indicates that the signals of these two detections point, in consistent directions, to conflicting levels of the same Fc species. This would cause the model to infer quantities of those contested Fc species that best accommodate the two detections if both of those detections are fit during optimization; but, if one conflicting detection were left out, the model could infer Fc species abundances that agree with the included detection without prediction penalties from the left-

**Table 1. Binding affinities ($K_a$ in units of $M^{-1}$) between each antibody Fc species and detection [22].**

| | IgG1 | IgG1f | IgG3 | IgG3f |
|---|---|---|---|---|
| **FcγRIIA-131H** | 5.20E6 | 5.20E6 | 8.90E5 | 8.90E5 |
| **FcγRIIA-131R** | 3.50E6 | 3.50E6 | 9.10E5 | 9.10E5 |
| **FcγRIIB** | 1.20E5 | 1.20E5 | 1.70E5 | 1.70E5 |
| **FcγRIIIA-158V** | 2.01E6 | 0 | 9.80E6 | 0 |
| **FcγRIIIA-158F** | 1.17E6 | 0 | 7.74E6 | 0 |
| **FcγRIIIB** | 2.00E5 | 0 | 1.10E6 | 0 |

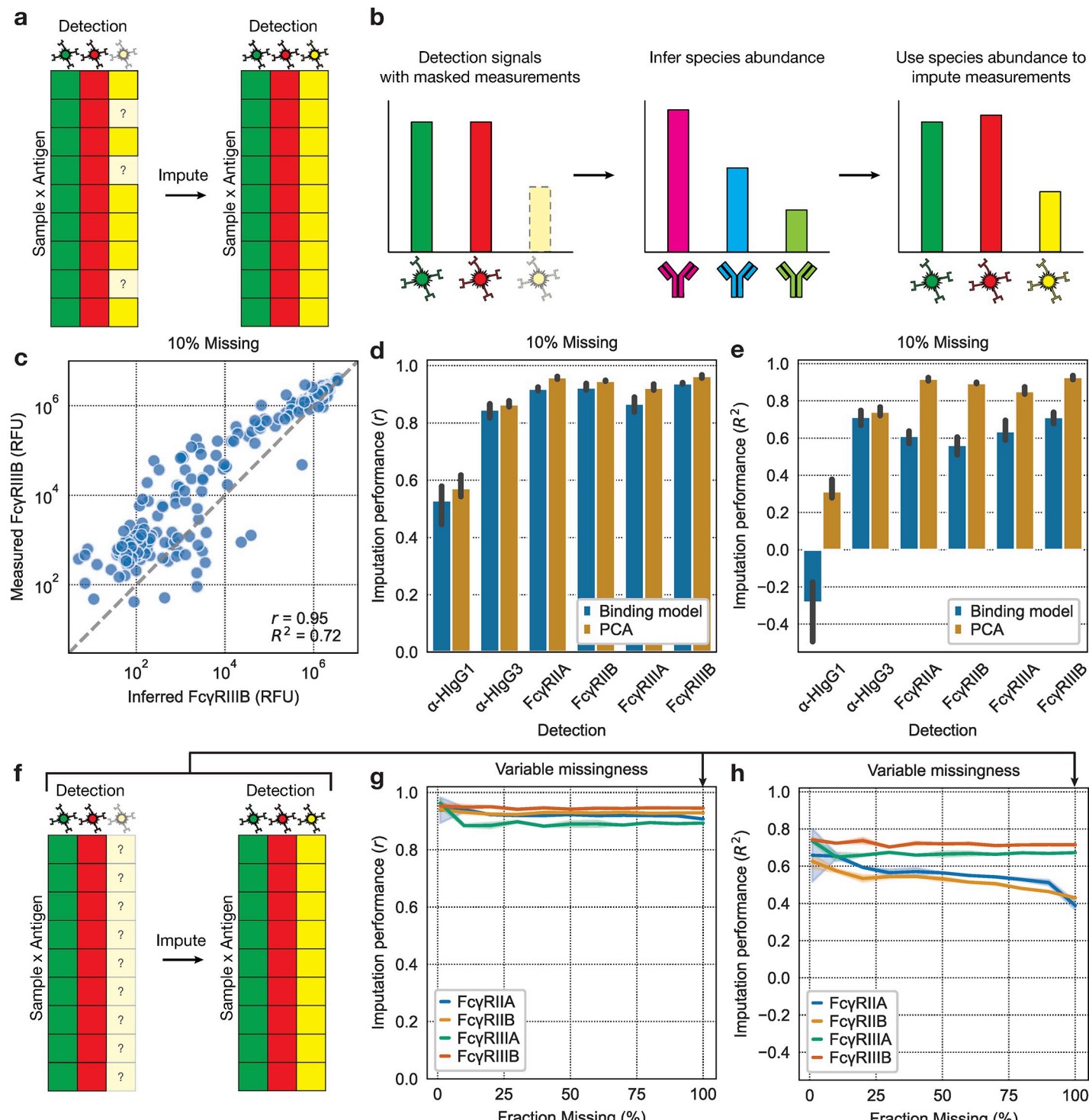

**Fig 3. The binding model effectively imputes unseen measurements. (a)** To measure the binding model's imputation performance, we start with a real systems serology dataset and then mask measurements corresponding to a particular detection and use the model to impute them. **(b)** The model imputes detection signals by using the incomplete data to infer the antibody abundances, which are then used to infer the left-out signals. **(c)** Imputed versus actual measurements. The metrics shown on the plot relate the $\log_{10}$ of each of the plotted values. 'r' signifies the Pearson correlation and '$R^2$' signifies the coefficient of determination. **(d)** Pearson correlation and **(e)** coefficient of determination between actual and imputed values for binding model and PCA when 10% of values are dropped. **(f)** Dataset schematic for imputation at 100% missingness for a single detection. **(g)** Pearson correlation and **(h)** coefficient of determination between actual and imputed values for the binding model at various percentages of missing values.

out detection. This would then cause a consistent bias in the model's imputations of the left-out detection.

Note that we don't examine the imputation accuracy of the subclass detections, α-hIgG1 and α-hIgG3, at variable percentages of missing values. We found that in some studies, such as this one, different concentrations of serum were used for each of these detections [11,14], preventing use of a binding model since the antibody compositions on the beads would almost certainly differ. Instead, we decided to model these detections with logistic curves that are fit on a dataset-by-dataset basis. This flexible modeling approach prompted by uncertainty in experimental conditions means that the subclass detection parameters will be determined by their specific experimental conditions, and thus cannot be compared across datasets. This is not a severe limitation, however, because there are relatively few such dataset-specific parameters (so they can be determined from a small number of samples) and because the Fc species quantities, which is what we're ultimately after, represent biological qualities of only the sample (i.e., no batch effects), and thus can be compared across datasets.

The binding model can also infer the number of antibodies with various glycan modifications, such as fucosylation. The presence of fucose on the Asn-297 N-linked glycan of IgG significantly reduces its binding affinity to FcγRIIIA, but has little effect on the affinity to other FcRs, such as FcγRIIA (Fig 4A) [22,24,25]. This suppresses IgG effector responses, particularly antibody-dependent cellular cytotoxicity (ADCC) by natural killer (NK) cells, and so is of particular interest in antibody therapeutics [24,26]. Because the reduction in binding affinity to FcγRIIIA caused by IgG fucosylation is well-characterized, we can separate fucosylated and afucosylated IgG as different Fc species to the model and quantify them (Fig 4B). One explanation for the model's capability to infer fucose is that Fc fucosylation is expected to have significant and isolated effects on the FcγRIII signal in the systems serology assay, which is distinct from the effects of other types of Fc variation. For example, changes in subclasses are generally expected to affect the signals for a broader set of detections, given their broad binding affinity differences to the detections relative to changes in fucosylation.

We sought to validate our model's IgG fucosylation inferences by comparing them to real, experimental measurements of both detection signals and effector functions to verify the presence of expected mechanistic relationships. In the SARS-CoV-2 data [11], we see a negative correlation between inferred IgG fucosylation and the FcγRIIIA to FcγRIIA signal ratio (Fig 4C), as well as a positive correlation between the abundance of afucosylated IgG and antibody-dependent natural killer cell activation (a proxy for ADCC) (Fig 4D and 4E), as expected. These relationships indirectly support our IgG fucosylation inferences.

We claim that the binding model infers antibody Fc features that better explain downstream effector response than existing antibody Fc characterizations. One reasonable prediction based on this claim is that effector responses are more accurately predicted by the antibody Fc features predicted by our model than they are by antibody Fc measurements that are already commonly available in systems serology assays. The class of commonly occurring systems serology measurements that most directly characterizes antibody Fc features is the set of IgG subclass detections (and other isotype detections). Indeed, other groups have previously attempted to relate these subclass detection measurements to effector response measurements [18], for example by using linear regression with the subclass detection measurements as regressors and the effector function measurements as response variables. In this same fashion, we benchmark the accuracy of regressing effector response against our Fc species inferences or, alternatively, subclass measurements (Fig 4F and 4G). We see that our model's inferences vastly outperform the subclass measurements in predicting effector responses. This prediction performance is likely attributable in large part to our model's separate inference of fucosylated and afucosylated IgG which, as previously stated, bind differentially to FcR.

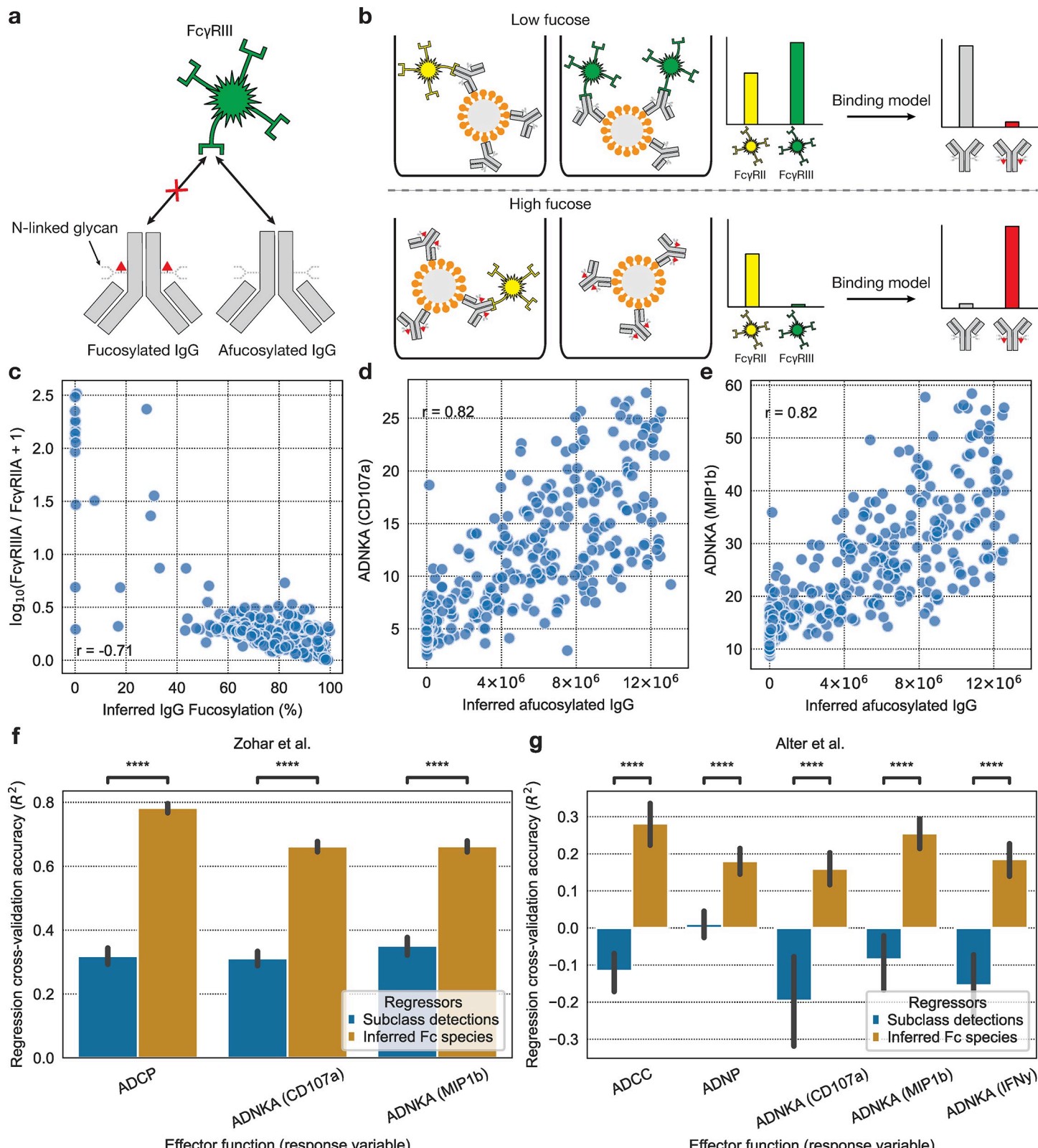

**Fig 4. Binding model infers IgG fucosylation, improving prediction of downstream effector response.** (a) IgG fucosylation blocks binding to FcγRIIIA with little effect on binding to the other FcRs, such as FcγRIIA. (b) Higher fucosylation of bead-bound antibodies leads to a lower FcγRIIIA signal relative to FcγRIIA signal. The binding model uses this information to infer how many antibodies are fucosylated. (c) Inferred IgG fucosylation versus measured FcγRIIIA signal / FcγRIIA signal. (d), (e)

Inferred abundance of afucosylated IgG versus antibody-dependent natural killer cell activation (ADNKA), measured by two markers: **(d)** CD107a and **(e)** MIP1b. **(f)**, **(g)** Effector function measurements were predicted with two sets of regressors: binding model-predicted abundances of antibody Fc species IgG1, IgG1f (fucosylated IgG1), IgG3, IgG3f and subclass detection measurements (with the set of subclasses varying by dataset). Repeated 8-fold cross validation with 10 repeats was used, and the coefficient of determination ($R^2$) on the validation sets is shown on the y-axis. **(f)** The regression performance for each set of regressors for the Zohar et al. SARS-CoV-2 data [11]. In this dataset, the available subclass detections used as regressors were α-hIgG1 and α-hIgG3 IgG. **(g)** The regression performance for the Alter et al. HIV dataset [10]. The available subclass detections used as regressors were α-hIgG1, α-hIgG2, α-hIgG3, and α-hIgG4 IgG. The Mann–Whitney $U$-test was used to define differences and the Benjamini–Hochberg method was used to adjust for multiple comparisons, with an adjusted $P(P_{adj})$ : $****P \leq 1 \times 10^{-4}$; $***P \leq 1 \times 10^{-3}$; $**P \leq 0.01$; $*P \leq 0.05$.

One previous systems serology study directly measured fucosylation by capillary electrophoresis (CE). While at first this seemed like an opportunity to directly validate our fucosylation inferences, the CE measurements were inconsistent with both the detection binding data and effector function measurements, suggesting challenges in accurately measuring glycan features in this manner (S5 Fig and S2 Text).

## Binding model reveals novel patterns of IgG fucosylation in SARS-CoV-2 and HIV infection

We next applied the model with the aim of discovering new patterns in natural antibody responses. While the model is generic to any antibody Fc species and detection for which binding affinities are known, we focus on using the model to infer antibody Fc fucosylation because (1) Fc fucosylation is difficult and expensive to measure directly, (2) we possess some prior knowledge about Fc fucosylation which allows us to further validate our model on real systems serology data, and (3) because Fc fucosylation causes well-characterized and sparse effects on Fc-FcR affinity and this simplicity makes it a suitable first domain of model application.

We first analyzed the same systems serology data from SARS-CoV-2 patients [11]. We used the binding model to infer the percentage of fucosylated antibodies targeting each antigen (Fig 5A–5C). As this dataset, like many others, contained no antibody glycosylation measurements, these fucosylation inferences are new information made available by our model. Antibodies targeting the SARS-CoV-2 nucleocapsid protein (N) were inferred to be more fucosylated than antibodies targeting the spike protein (S) (Fig 5A). This relationship holds when considering antibodies targeting all the antigens associated with the spike protein, compared with those targeting the nucleocapsid protein (Fig 5B). Furthermore, antibodies targeting the S1 subunit of the spike protein are inferred to be more fucosylated than antibodies targeting the S2 subunit (Fig 5A). We also found that all the SARS-CoV-2 antigen responses were inferred to be less fucosylated in patients with acute respiratory distress syndrome (ARDS) compared to patients without ARDS (Fig 5C).

We further explored the relevance of antibody fucosylation in subjects who received the ChAdOx1 SARS-CoV-2 (AZD1222) vaccine [14]. This data was collected 2 weeks after the booster, and each subject was followed up with later to determine whether they tested positive for SARS-CoV-2 at any point since vaccination. Vaccine protection was correlated with inferred afucosylation of antibodies targeting spike protein antigens from the wild type (WT) and beta variant SARS-CoV-2 (Fig 5D).

As an application to data with more complex antibody compositions, we examined systems serology data previously collected in a separate study from HIV patients. This study includes CE IgG fucosylation measurements for the gp120.SF162 HIV antigen [10]; we used our model to infer IgG fucosylation for the remaining antigens. We first analyzed these anti-HIV IgG fucosylation inferences across four patient categories: elite controllers, who suppressed the virus to the extent that there was no longer evidence of viremia; viremic controllers, who suppressed the virus to barely transmissible levels; untreated progressors, who were chronically

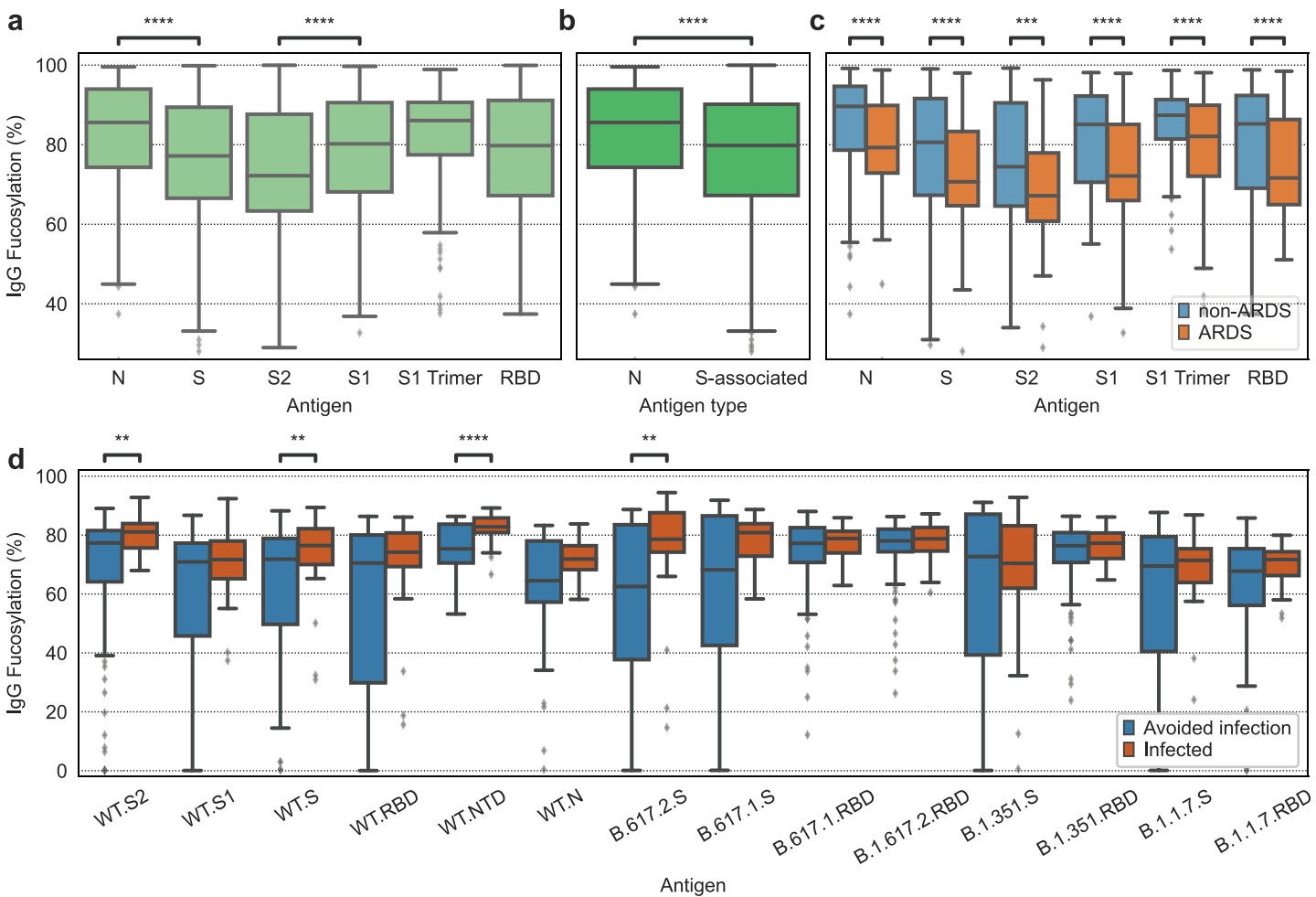

**Fig 5. In COVID-19, inferred IgG fucosylation varies by target antigen, symptom severity, and vaccine efficacy.** Inferred fucosylation of IgG by **(a)** target antigen, **(b)** target antigen type, **(c)** target antigen and presence of ARDS. **(d)** SARS-CoV-2 vaccine protection is associated with inferred afucosylation of IgG targeting spike protein antigens. The Mann–Whitney $U$-test was used to define differences and the Benjamini–Hochberg method was used to adjust for multiple comparisons, with an adjusted $P(P_{adj})$ : $****P \leq 1 \times 10^{-4}$; $***P \leq 1 \times 10^{-3}$; $**P \leq 0.01$; $*P \leq 0.05$.

progressive; and treated progressors, who were also chronically progressive, but were treated with anti-retroviral therapy and thus had no evidence of viremia (Fig 6A). We see large variation in antibody fucosylation across both antigens and subject classes. In the hopes of isolating differences relevant to effective immune response, we compared fucosylation between elite controllers and the other groups (Fig 6B). Elite controllers possess IgG that is more fucosylated against most antigens, particularly gp120.Du151, gp120.Du156.12, and gp120.ZM109F. This correlation between HIV infection severity and IgG afucosylation is like the correlation we found between COVID-19 severity and IgG afucosylation.

There are also significant differences in antibody fucosylation across antigens, most prominently between pr55$^{Gag}$ and the others (Fig 6A). We investigated these differences across all subjects directly (Fig 6C) by grouping the antigens into three categories: pr55$^{Gag}$ is a polyprotein containing units for intra-envelope structures, Env is the envelope spike protein, and p24 forms the viral capsid. IgG targeting pr55$^{Gag}$ is indeed more fucosylated than IgG targeting the other two groups of antigens. We also looked at the relationship between Fc afucosylation and HIV severity across these antigen groups (Fig 6D). From this, we extend the observations

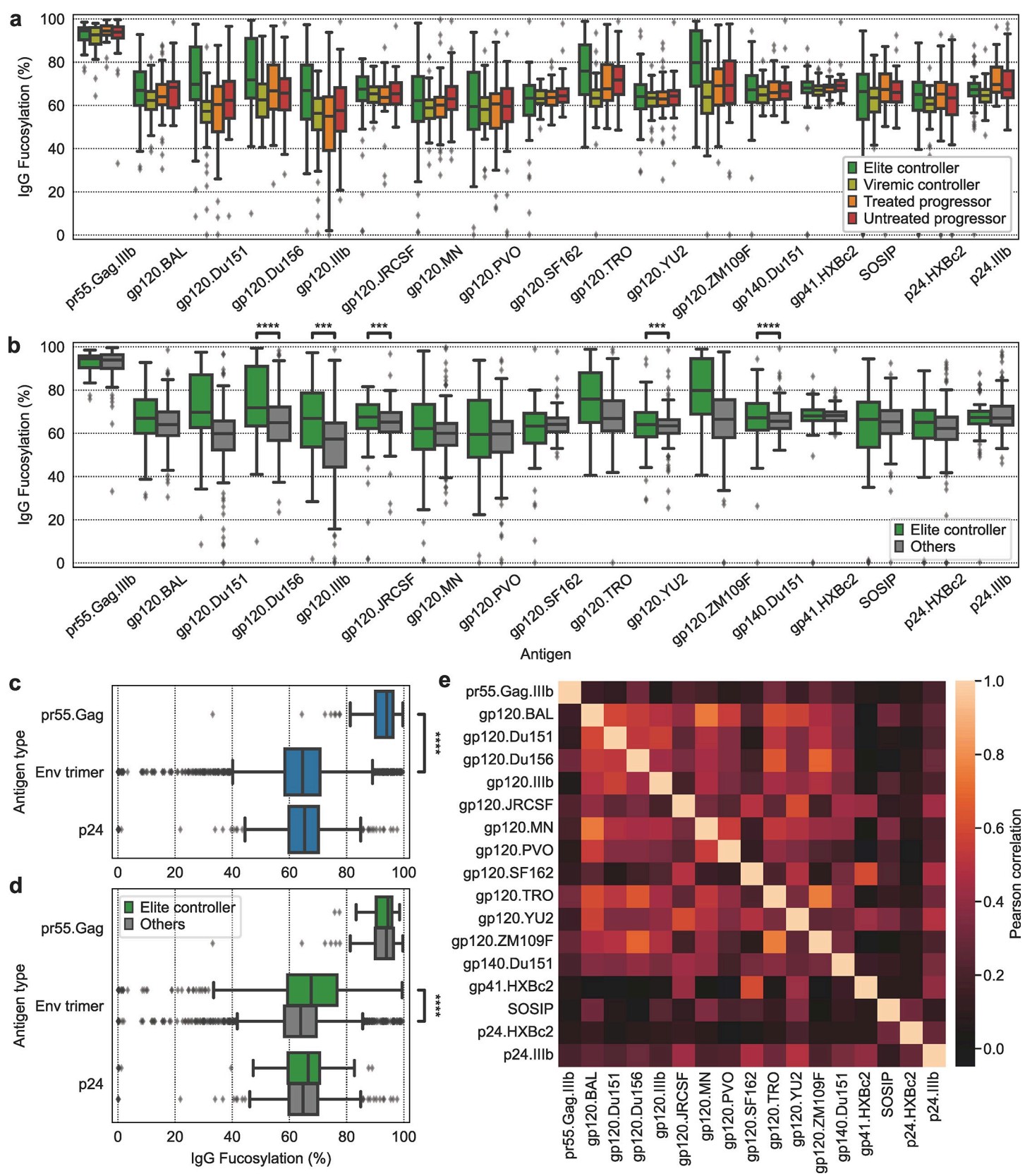

**Fig 6. Inferred IgG fucosylation correlates with HIV severity and is lower for membrane-associated antigens. (a), (b)** Inferred IgG fucosylation by target antigen and patient status for HIV-infected subjects. **(c)** Inferred IgG fucosylation by antigen type. **(d)** Inferred IgG fucosylation by antigen type and patient status. **(e)** IgG fucosylation was inferred for each sample and antigen. The fucosylation inferences for each antigen were compared across samples and used to compute a Pearson correlation coefficient. The pairwise correlation in IgG fucosylation between each antigen is shown. A Mann–Whitney $U$-test was used to define differences and the Benjamini–Hochberg method was used to adjust for multiple comparisons, with an adjusted $P (P_{adj})$ : ****$P \leq 1 \times 10^{-4}$; ***$P \leq 1 \times 10^{-3}$; **$P \leq 0.01$; *$P \leq 0.05$.

made in Fig 6B; the relationship between Fc afucosylation and HIV severity is exclusive to Env trimer antigens.

We also sought to understand the variability in IgG fucosylation by antigen. Because techniques for directly measuring IgG fucosylation, such as CE or LC-MS, have lower throughput than measurement of detection binding, measurements of IgG fucosylation, when available, are typically only collected for at most one or two antigens [10,27], in contrast to the large panels of antigens against which multiplexed binding is measured. We wanted to understand to what degree the fucosylation of IgG targeting one antigen is representative of the fucosylation of IgG targeting other antigens associated with the same pathogen. This would help determine how comprehensive these direct measurements of IgG fucosylation are. We found a high degree of variation in IgG fucosylation by antigen across subjects (Fig 6E), indicating that antigen-specific IgG fucosylation cannot be inferred using the IgG fucosylation for another antigen from the same pathogen. This difference is most pronounced between antigens from different structural units of the virus but is also present between antigens from the same structural unit and different viral strains.

## Binding model enables the optimization of future systems serology assays

We also used the model to explore how systems serology protocols might be reduced in experimental workload while capturing the same information. Recall that one of our model validation strategies was to impute missing detection signals in systems serology data, even when that detection is completely absent in the dataset. We can repurpose this mechanism to infer the signal of a detection which was never present in the dataset and thus theoretically forgo the need to collect that measurement.

We sought to find a minimal subset of detections from which the signals of all other detections of interest can be inferred. α-hIgG1 IgG and α-hIgG3 IgG cannot be inferred at 100% missingness due to their learned parameters, so we always included these in the subset of measurements seen by the model. Using the SARS-CoV-2 dataset [11], we explored all possible pairwise combinations of FcRs. For each pair, we dropped their corresponding measurements and measured the model's capability to impute these quantities (Fig 7A, 7C and 7E–7H). As we saw previously (Fig 3), the Pearson correlation was often higher than the coefficient of determination, indicating a bias. In practice, this bias might not be a severe issue, as relative differences between subjects and antigens are still preserved. Under these criteria, the model can impute most of these combinations quite well. The imputation performance, measured by the Pearson correlation, is notably worse for (FcγRIIA, FcγRIIB) and (FcγRIIIA, FcγRIIIB). Because FcγRIIIA and FcγRIIIB are fucosylation-responsive while FcγRIIA and FcγRIIB are not, these results indicate that accurate inference requires the inclusion of at least one fucosylation-responsive FcR and one fucosylation-unresponsive FcR.

We then hoped to minimize the measured subset of detections further by dropping three FcRs at a time (Fig 7B and 7D). While this generally leads to lower imputation performance than with two FcRs, the combination of (FcγRIIB, FcγRIIIA, FcγRIIIB) can still be imputed moderately well. In this scenario, excluding these detections would reduce the number of detections from six (α-hIgG1 IgG, α-hIgG3 IgG, FcγRIIA, FcγRIIB, FcγRIIIA, FcγRIIIB) to three (α-hIgG1 IgG, α-hIgG3 IgG, FcγRIIB).

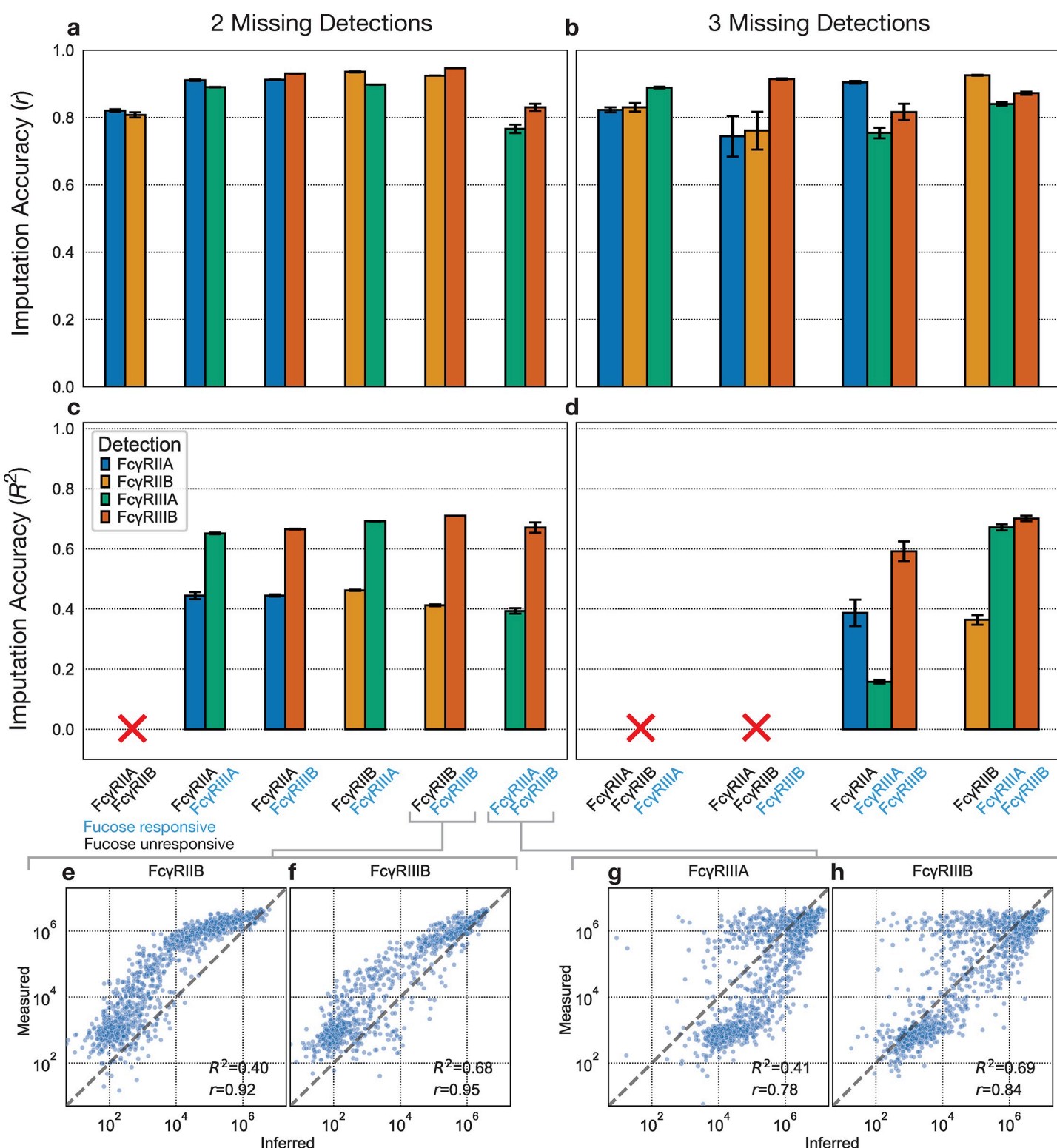

**Fig 7. Identifying a minimal set of detections.** Combinations of two FcRs are dropped, and their signals are imputed. The **(a)** Pearson correlation and **(c)** coefficient of determination between the $\log_{10}$ of the imputed and measured values. **(e), (f)** Imputed vs measured values for FcγRIIB and FcγRIIIB respectively, when they are left out together. 'r' signifies the Pearson correlation and 'R²' signifies the coefficient of determination. **(g), (h)** Imputed vs measured values for FcγRIIIA and FcγRIIIB respectively, when they are left out together. **(b) (d)** The same metrics as **(a)** and **(c)** when three FcRs are imputed. An 'x' indicates that one of the specified values exceeds the lower y-axis limit.

## Discussion

Here, we show that a mechanistic binding model complements systems serology by quantifying Fc structural features of antibodies, and by defining which types of measurements are most informative. This approach models the microscopic binding mechanics in the assay itself (Fig 1) and uses this fact to infer the abundances of each antibody Fc species. While the model's antibody inferences are difficult to validate directly, we used several approaches to indirectly support their accuracy (Figs 2, 3, and 4). With this capability, we then characterized antibody-mediated immune responses better than could be done using the existing systems serology data alone—specifically, by inferring the proportion of fucosylated antibodies (Figs 5 and 6). Notably, this characterization provides unique opportunities for engineering of antibody-based therapies by virtue of its direct description of antibody Fc structural features, rather than only the immune interaction profiles which arise from them (Fig 5D). Finally, we use the model to describe the information content of different detections typically used in systems serology and discover optimizations that could be used to make future systems serology assays more efficient (Fig 7).

The antibody Fc information that we obtained with the binding model aligns with and extends existing understanding of antibody-mediated protection against SARS-CoV-2. The model's inferences of IgG fucosylation are particularly useful because these quantities are more difficult to measure than other properties such as subclass composition. Our fucosylation inferences are further validated by their recapitulation of previously measured fucosylation patterns. For one, the larger degree of fucosylation with N-targeting IgG relative to S-targeting IgG is consistent with a previously established relationship that membrane-associated antigens elicit IgG with lower fucosylation [27]. Additionally, the lower degree of S-targeting IgG fucosylation among patients with ARDS compared to those without ARDS replicates a previous finding [27]. We also discover new relationships based on our model's fucosylation inferences. For one, we extend the ARDS relationship from S-targeting IgG to IgG targeting other SARS-CoV-2 antigens. We also find that S2-targeting IgG is less fucosylated than S1-targeting IgG. As these proteins form subunits of the same protein (S) on the viral surface, this difference in fucosylation could lend specificity to the theory that membrane-associated antigens induce afucosylated IgG. S1 contains the receptor-binding domain (RBD), which recognizes and binds to the angiotensin-converting enzyme 2 (ACE2) receptor on the host cell membrane [28]. Subsequently, S2 mediates viral fusion with the host cell membrane and remains embedded in the host membrane after fusion [28–30]. S1, on the other hand, commonly dissociates after viral fusion [29,30]. Therefore, S2 is likely more abundant on the host cell membrane than S1 [29,30]. Given this background, our results may point toward the significance of the host context in the relationship between membrane antigens and afucosylated antibody production, whereby afucosylated IgG production is stimulated by co-signaling between B cell and antigen-presenting host cell in parallel to B cell receptor (BCR) activation, as originally proposed by Larsen et al. [27]. Our inferences also indicate that COVID-19 vaccination is associated with the induction of afucosylated S-targeting IgG (Fig 5D). This finding explains previously observed differences in antibody effector profiles between protected and unprotected vaccinees. Namely, in the study that we extracted this IgG fucosylation information from, the authors observed that FcγRIIIB binding was highly predictive of vaccine protection [14]. Our finding further emphasizes the need for an understanding of how to tune IgG glycosylation in vaccination [31,32].

We also gained IgG fucosylation information for an unprecedented breadth of HIV antigens, which expands and questions our understanding of antibody-mediated protection against HIV. We found that IgG targeting a range of HIV antigens were more fucosylated for

ECs (Fig 6A and 6B). This could arise from a similar mechanism as that which distinguishes COVID-19 patients with ARDS; namely, a more advanced infection results in the production of afucosylated IgG. However, this relationship does not appear for all antigens, and the underlying determinant for those antigens with IgG fucosylation differences across disease severity is not entirely clear. These antigens against which IgG fucosylation depends on disease severity seem to be limited to the Env trimer (Fig 6D), but not all Env trimer antigens have this property, such as gp120.SF162 (Fig 6B). Further investigation may be warranted to explain these differences. We also see a significant degree of independence of IgG fucosylation across HIV antigen types (Fig 6E). This independence may help explain the apparent contrast of our observed relationship between IgG afucosylation and HIV severity with a previous finding where HIV-specific IgG afucosylation was greater for controllers than progressors [33]. It is plausible that the single HIV antigen they examined had little correlation to other relevant HIV antigens, such as the ones we examined here. On a related note, this independence of IgG fucosylation across antigens emphasizes the value of our model's ability to infer IgG fucosylation for many antigens, as these results are ultimately distinct from an IgG fucosylation analysis for one or two antigens offered by CE or LC-MS. We also discovered a relationship between IgG fucosylation and antigen context in HIV (Fig 6C) which largely resembles what we found for COVID-19; afucosylated IgG is produced against membrane-associated antigens. pr55$^{Gag}$ is a polyprotein containing units for intra-envelope structures [34], while Env exists on the viral envelope [35], and thus the increased fucosylation of IgG targeting pr55$^{Gag}$ reflects this relationship. However, it is somewhat surprising that IgG targeting p24 are afucosylated, as this is a component of the viral capsid and thus is not expected to be found on the host cell membrane or viral envelope. Our finding indicates that the p24 antigen is vulnerable to antibody targeting on the viral or host cell membrane. This aligns with previous findings that p24 is expressed on a fraction of HIV-infected T cells and that anti-p24 antibodies effectively halt viral replication in vitro [36].

This quantitative analysis of systems serology brings our attention to measurement redundancies commonly found in these assays. On one hand, by profiling the information content of commonly used detections from the assay, we found that certain detections could be omitted from future experiments. Ultimately, the overlap in information content between measurements arises from similar profiles of antibody binding affinities between detections [22]. On the other hand, Fc species with similar binding affinities to the detections are difficult to resolve even with all measurements, suggesting the need for new detection reagents to resolve these Fc species. In either case, the model can help to reason about what conclusions can be drawn given the detections and Fc species present.

We envision several other directions in which this approach could be taken. While we analyzed IgG fucosylation extensively, other variation in antibody glycosylation, such as galactosylation and sialylation, have previously been implicated in a range of diseases [33,37–43]. Because it is difficult to directly measure glycosylation, and because we lack detections which can quantify these features in a deconvoluted manner, analysis in this area is lacking. The binding model could help quantify these features. There are also opportunities to solidify and expand the inferences made available by the model. First, the quantitative effect of IgG glycosylation on binding affinity is not well-known mainly because it is difficult to create pure solutions of specific antibody glycoforms [22,44]. Enhanced estimates of these affinities could refine the model's inferences. Second, the binding model would become more effective with increased homogeneity in systems serology protocols both between detections and studies. For example, as mentioned earlier, we chose to use a more flexible modeling approach for the subclass detections because of the varying serum concentrations used with them. Finally, other data modalities from systems serology, such as effector function and glycosylation measurements, could also be integrated into this approach, for example, by modeling FcR crosslinking

and effector cell activation [19] or by constraining the model's antibody inferences based on known Fc glycosylation compositions.

Antibody effector functions are an important aspect of immunity. Systems serology can measure immune receptor engagement one step upstream of effector responses, as well as effector responses directly, at an unparalleled breadth and throughput. An important next step is a mechanistic understanding of how effector responses directly relate to antibody Fc features as well as how the immune system naturally chooses these Fc features to control effector response in vivo. This understanding could help us artificially program effector responses that have shown success in providing natural immunity. However, this understanding requires advancements in our capability to profile effector-relevant antibody Fc features. The application of a binding model to systems serology that we present here is one approach to improving this capability.

## Methods

### Multivalent binding model for systems serology

We start by considering beads coated with a single antigen species. These beads are mixed with patient serum containing antibodies, and the antibodies bind to the antigen on the bead via their Fab region. We assume that the antibodies and beads are mixed homogenously such that each bead is bound by the same composition of $N_R$ antibody Fc species, $R_1, R_2, \ldots, R_{N_R}$ with abundances $R_{tot,1}, R_{tot,2}, \ldots, R_{tot,N_R} = \overrightarrow{R_{tot}}$. These bead-antigen-antibody complexes are separated into distinct wells, one for each type of detection. In well $W_i$, multivalent detection $L_i$ of valency f is introduced to this well at a concentration of $L_0$. At steady state, the total number of detection molecules bound to the bead, $L_{boound,i}$, can be expressed as [45]:

$$L_{bound,i} = \frac{L_0}{K_x^*} \left[ \left( 1 + \sum_{j=1}^{N_R} R_{eq,j} K_{a,ij} K_x^* \right)^f - 1 \right] \tag{1}$$

Where $K_x^*$ is the crosslinking constant, which quantifies the difference between free monovalent binding and binding of a subunit which is part of a multivalent complex that already has another bound subunit. This constant thus captures steric effects and local receptor clustering. $K_{a,ij}$ is the binding affinity between species $R_j$ and detection $L_i$. $R_{eq,j}$ is the number of unbound antibodies of species $R_j$ at equilibrium. Our objective is to express $L_{bound,i}$ in terms of $\overrightarrow{R_{tot}}$, rather than $R_{eq,1}, R_{eq,2}, \ldots, R_{eq,N_R} = \overrightarrow{R_{eq}}$. Therefore, we seek a relationship between $\overrightarrow{R_{tot}}$ and $\overrightarrow{R_{eq}}$. In fact, it suffices to find $\Phi_i \equiv \sum_{j=1}^{N_R} R_{eq,j} K_{a,ij} K_x^*$ in terms of $\overrightarrow{R_{tot}}$. We also define $\varphi_{i,j} \equiv R_{eq,j} K_{a,ij} K_x^*$. With these new variables defined, Eq (1) becomes

$$L_{bound,i} = \frac{L_0}{K_x^*} \left[ (1 + \Phi_i)^f - 1 \right]. \tag{2}$$

We use conservation of mass:

$$R_{tot,j} = R_{eq,j} + R_{bound,j}, \tag{3}$$

where $R_{bound,j}$ is the number of antibodies of species $R_j$ bound to detection at equilibrium. We can express $R_{bound,j}$ in terms of $R_{eq,j}$ as [45]

$$R_{bound,j} = \frac{L_0 f}{K_x^*} \varphi_{ij} (1 + \Phi_i)^{f-1}.$$

Inserting this into Eq ([2]), specifying $R_{eq,j}$ in terms of $\varphi_{ij}$, and rearranging, we have:

$$R_{tot,j} = R_{eq,j} + \frac{L_0 f}{K_x^*} \varphi_{ij} (1 + \Phi_i)^{f-1}$$

$$R_{tot,j} = \frac{\varphi_{ij}}{K_{a,ij} K_x^*} + \frac{L_0 f}{K_x^*} \varphi_{ij} (1 + \Phi_i)^{f-1}$$

$$R_{tot,j} K_x^* = \frac{\varphi_{ij}}{K_{a,ij}} + L_0 f \varphi_{ij} (1 + \Phi_i)^{f-1}$$

$$R_{tot,j} K_x^* = \varphi_{ij} \left[ \frac{1}{K_{a,ij}} + L_0 f (1 + \Phi_i)^{f-1} \right]$$

$$\varphi_{ij} = \frac{R_{tot,j} K_x^*}{\frac{1}{K_{a,ij}} + L_0 f (1 + \Phi_i)^{f-1}}$$

Replacing $\varphi_{ij}$ with $\Phi_i$ by summing over j:

$$\sum_{j=1}^{N_R} \varphi_{ij} = \sum_{j=1}^{N_R} \frac{R_{tot,j} K_x^*}{\frac{1}{K_{a,ij}} + L_0 f (1 + \Phi_i)^{f-1}}$$

$$\Phi_i = \sum_{j=1}^{N_R} \frac{R_{tot,j} K_x^*}{\frac{1}{K_{a,ij}} + L_0 f (1 + \Phi_i)^{f-1}}. \tag{4}$$

We solve for $\Phi_i$ in Eq ([3]) using the Newton-Raphson method as implemented in SciPy.

Equipped with these relationships, we can compute $L_{bound,i}$ from $\overrightarrow{R_{tot}}$ by first solving for $\Phi_i$ in Eq ([4]) and then plugging this into Eq ([2]). We have focused on just one well and detection, but this process is repeated in $N_L$ wells, each of which contains beads from the same original mixture, and thus beads with identical bead-bound antibody abundances, $\overrightarrow{R_{tot}}$. We can compute the number of bound molecules of each detection type and combine them into a vector: $L_{bound,1}, L_{bound,2}, \ldots, L_{bound,N_L} = \overrightarrow{L_{bound}}$. This entire process implements the function $f_{L \leftarrow R}$ : $\mathbb{R}^{N_R} \rightarrow \mathbb{R}^{N_L}$ where $\overrightarrow{L_{bound}} = f_{L \leftarrow R}(\overrightarrow{R_{tot}})$. $f_{L \leftarrow R}$ is also referred to as the antibody-to-signal function.

Each detection molecule is fluorescently tagged, and this fluorescence is measured for each well, yielding a vector $\overrightarrow{\mathcal{L}} \in \mathbb{R}^{N_L}$. If we assume that each detection molecule contributes equally to the fluorescence, i.e., the detection abundance and fluorescence are linearly related as $\overrightarrow{L_{bound}} = \gamma \overrightarrow{\mathcal{L}}$, for some constant $\gamma$, then the fluorescence measurements are also indirect measurements of the number of detection molecules bound to the bead. Therefore, with $\overrightarrow{L_{bound}}$ available from the assay data, we seek a way to compute $\overrightarrow{R_{tot}}$ as $\overrightarrow{R_{tot}} = f_{R \leftarrow L}(\overrightarrow{L_{bound}})$ where $f_{R \leftarrow L} = f_{L \leftarrow R}^{-1}$. $f_{R \leftarrow L}$ is also referred to as the signal-to-antibody function.

To implement $f_{L\leftarrow R}^{-1}$, we use numerical optimization. Namely, we solve

$$\underset{\overrightarrow{R_{tot}}}{argmin} \left\| \log_{10} f_{L\leftarrow R}(\overrightarrow{R_{tot}}) - \log_{10} \overrightarrow{L_{bound}} \right\|^2$$

$$= \underset{\overrightarrow{R_{tot}}}{argmin} \left\| \log_{10} f_{L\leftarrow R}(\overrightarrow{R_{tot}}) - \log_{10} \gamma \overrightarrow{\mathcal{L}} \right\|^2. \tag{5}$$

In other words, we are finding the vector $\overrightarrow{R_{tot}^*}$ that minimizes the squared residuals between $\log_{10} f_{L\leftarrow R}(\overrightarrow{R_{tot}})$ and $\log_{10} \gamma \overrightarrow{\mathcal{L}}$, where the log is added because the data spans many orders of magnitude.

At first, we attempted to fit $\gamma$, but this made the fitting too flexible, resulting in unrealistic values of $\overrightarrow{R_{tot}}$. Specifically, we would sometimes see that the average inferred number of antibodies bound to each bead was many orders of magnitude larger than we would realistically assume. To avoid this issue, we chose to set $\gamma = 1$, in which case the measured fluorescence is the detection abundance. While this assumption is almost certainly inaccurate, we surmised that it was acceptable after validating the model. Future assays to which this model is applied would be benefited by a calibration step, where the mean fluorescence per detection molecule is measured, thus providing $\gamma$.

Eq (5) describes the assay and fitting process for one serum sample and one antigen, where there are $N_L$ fluorescence measurements. In the assay, these experimental steps are conducted for $N_S$ samples and $N_A$ antigens, resulting in $N_S \times N_A \times N_L$ total fluorescence measurements, $N_L$ measurements for each sample and antigen. This model performs the same computation on each sample and antigen, and thus it is natural to arrange the input data into a matrix of shape $(N_S \times N_A, N_L)$. The model then computes Eq (5) for each row of the input, and outputs the antibody abundances for each sample and antigen, a matrix of shape $(N_S \times N_A, N_R)$. We use the least_squares function implemented in the SciPy python package for optimization.

One additional consideration is that in some systems serology assays, a different set of dilutions was used for each subclass quantification [11]. Due to differences in antibody-antigen affinities, this likely leads to differing bound antibody compositions. Consequently, the assumptions of the binding model would not hold for these experiments. To account for this in the model, we opted to change $f_{L\leftarrow R}$ to be more flexible for the subclass-specific detections by using a 4-parameter logistic curve, in effect restricting that the order of the subclass abundances across samples is preserved. Specifically,

$$\overrightarrow{L_{bound,i}} = \beta_{i,1} + \frac{\beta_{i,0} - \beta_{i,1}}{1 + \left(\frac{\sum_{j \in I_i} R_{tot,j}}{\beta_{i,2}}\right)^{\beta_{i,3}}},$$

where $\mathbf{I_i}$ is the set of indices that correspond to antibodies that detection $L_i$ selectively binds to. We assume that a subclass detection binds all glycoforms of its target subclass and no other species, so if $L_i$ were an $\alpha$-hIgG1 detection, then $\{R_{tot,j} | j \in \mathbf{I_i}\}$ would be all of the glycoforms of hIgG1. The parameters of the logistic curve, $\beta_{i,0}, \ldots, \beta_{i,3}$ are fit along with the antibody abundances. Unlike the antibody abundances, the logistic curve parameters are shared across samples and antigens. Therefore, in our fitting process, we minimize the sum of the squared residual term specified in (5), which we call $\Gamma$, for all samples and antigens simultaneously.

We regularize the logistic curve slope parameters (i.e., Hill coefficients), $\beta_{i,3}$, to penalize deviations above 1 based on the assumption that there is minimal cooperative binding between

**Table 2. Software tools.**

| Source | Reference | Identifier |
|---|---|---|
| NumPy Python package | https://numpy.org/ | v1.24.3 |
| Xarray Python package | https://xarray.dev/ | v2022.12.0 |
| SciPy Python package | https://www.scipy.org/ | v1.10.1 |
| Pandas Python package | https://pandas.pydata.org/ | v1.5.3 |
| Seaborn Python package | https://seaborn.pydata.org/ | v0.11.2 |
| scikit-learn Python package | https://scikit-learn.org/ | v1.2.2 |
| statsmodels Python package | https://www.statsmodels.org/ | v0.14.0 |
| statannotations Python package [46] | https://github.com/trevismd/statannotations | v0.5.0 |
| Python | https://www.python.org | v3.11.1 |

the subclass detections and their target species. This regularization is achieved by adding the following penalty term to Γ:

$$\sum_{i \in \mathbf{I}_\ell} \left( \frac{[\max(\beta_{i,3} - 1, 0)]^2 \times N_S \times N_A \times N_L}{|\mathbf{I}_\ell|} \right)^2$$

where $\mathbf{I}_\ell$ is the set of indices that correspond to ligands that are modeled with a logistic curve. The factor of $N_S \times N_A$ is added to ensure that the relative weight of this penalty term is invariant to the number of complexes being fit, and the factor of $N_L/|\mathbf{I}_\ell|$ is added to ensure that the relative weight of this term is invariant to the relative number of ligands fit with a logistic curve. We apply this strategy to all datasets analyzed in the paper.

All software tools that we used for the model implementation and analysis are listed in Table 2.

### Antibody Fc species selection

The chosen set of antibody Fc species must reflect the molecular features that lead to differential binding. We chose to exclude IgG2 and IgG4 because of their relatively low affinity to Fcγ receptors and low abundance in the analyzed studies. Fucosylated and afucosylated forms were considered because this glycan modification has substantial effects on FcγRIIIA interaction.

### Parameter selection

$K_{a,ij}$ is the matrix of binding affinities between each Fc species and detection (Table 1). Allelic variants are specified in the suffix for FcγRIIA and FcγRIIIA.

Fucosylation has been found to reduce the binding affinity of human IgG1 to FcγRIIIA and FcγRIIIB to undetectable levels [22]. $L_0$ is the concentration of each detection in solution which was defined as 1 µg/mL in the HIV study [10]. The other studies did not state the concentrations they used, and so we assumed a concentration 1 µg/mL. $K_x^*$, the crosslinking constant, was quantified in our previous work to be around $10^{-12}$ for immune complexes consisting of anti-TNP IgG bound to a BSA and TNP scaffold. We assumed that the difference in $K_x^*$ between these immune complexes and the multivalent detections formed with streptavidin tetramerization analyzed in this work was acceptable, and thus took $K_x^* = 10^{-12}$. $f$, the detection valency, was 4 for every Fcγ receptor in every dataset analyzed here.

### Datasets

We used three systems serology datasets in this work. The first contains measurements from COVID-19 patients [11], and we used the detection measurements α-hIgG1 IgG, α-IgG3 IgG,

FcγRIIA, FcγRIIB, FcγRIIIA, and FcγRIIIB. Because the allele of FcγRIIA and FcγRIIIA were not provided, we used the affinities for FcγRIIA-131H and FcγRIIIA-158V, respectively. This dataset was used for the imputation analysis, the comparison of inferred IgG fucosylation with FcγRIIIA / FcγRIIA and ADCC, and the analysis of SARS-CoV-2-targeting IgG fucosylation across patient features. The second dataset contains measurements from HIV patients [10], and we used the detection measurements α-hIgG1 IgG, α-hIgG3 IgG, FcγRIIA-131H, FcγRIIA-131R, FcγRIIB, FcγRIIIA-158V, FcγRIIIA-158F, and FcγRIIIB. This dataset was used for validating fucose inferences against CE and for analyzing IgG fucosylation across HIV antigens. The final dataset contains measurements from SARS-CoV-2 vaccine recipients [14], and we used the detection measurements α-hIgG1 IgG, α-hIgG3 IgG, FcγRIIA, FcγRIIB, FcγRIIIA, and FcγRIIIB. This dataset also did not include alleles for FcγRIIA and FcγRIIIA, and we handled this in the same way as we did the other SARS-CoV-2 dataset. We dropped all subjects in each dataset that contained any missing values for the detections we selected. Where the baseline for each detection and antigen pair was provided, we subtracted the baseline. If this subtraction resulted in negative values, we instead subtracted off the minimum measured value.

## Validation with synthetic data

We initialized random combinations of subclasses IgG1–4 by sampling from a $\log_{10}$ normal distribution with a $\log_{10}$-mean of 2 antibodies per complex and a $\log_{10}$-standard deviation of 0.4 antibodies per complex. We then inferred the signals of detections α-hIgG1 IgG, α-hIgG2 IgG, α-hIgG3 IgG, α-hIgG4 IgG, FcγRIIA-131R, FcγRIIB-232I, FcγRIIIA-158V, and FcγRIIIB using $f_{L \leftarrow R}$ as described above. We fitted the abundance of each species from these detection signals in $f_{R \leftarrow L}$ and compared these fitted abundances to the initial abundances. Noise was added as:

$$x_{noisy} = x \times \max[\mathcal{N}(\mu = 1, \sigma), 0]$$

$x$ contains either the detection abundances or the binding affinities. $\mathcal{N}$ indicates a Gaussian-distributed random number.

## Binding affinity sensitivity analysis

We initialized random combinations of subclasses IgG1–4 by sampling from the same distribution as above. We perturbed each binding affinity by moving it up 30% and down 30% separately and inferring the detection signals (as previously described) using that perturbed affinity and with the canonical values for the remaining affinities. These inferred detection signals were then used to infer the antibody abundances using the unperturbed affinities, which were compared with the original antibody abundances. The coefficient of determination was computed for each subclass separately using the original and inferred antibody abundances, and this was reported by subclass and affinity.

## Imputation

We inferred the abundance of each Fc species using $f_{R \leftarrow L}$ but hiding a portion of the values corresponding to one detection. For each sample and antigen for which the detection was hidden, we then use $f_{L \leftarrow R}$ to infer the missing detection abundance from the antibody abundances. We introduced missingness and imputed in the same way for PCA.

### Prediction of IgG fucosylation

To infer IgG fucosylation with the binding model, we first inferred the abundances of species IgG1, IgG1f, IgG3, and IgG3f. We then computed the fucosylation fraction as:

$$F = \frac{R_{tot,IgG1f} + R_{tot,IgG3f}}{R_{tot,IgG1} + R_{tot,IgG1f} + R_{tot,IgG3} + R_{tot,IgG3f}}$$

### Optimization of future assays

In this section we ran imputation tests in the same way as we described earlier, except that we hid all values from one or more detections. We then compared our model's imputation capability across combinations of detections.

## Supporting information

**S1 Text. Further explanation of S1 Fig.** Reasoning behind chosen distributions of synthetic Fc species abundances.
(DOCX)

**S2 Text. Further explanation of S5 Fig.** Comparison of fucosylation inferences from the binding model with fucosylation measurements from capillary electrophoresis.
(DOCX)

**S1 Fig. Synthetic antibody Fc species abundances are sampled from a distribution based on subclass-specific detection measurements from systems serology datasets. (a-h)** Distributions of measured signals (baseline-subtracted) for subclass-specific detections for each of the three datasets we used in this paper (rows) and each subclass (columns). The Kaplonek dataset did not have data for $\alpha$-IgG2 IgG. **(i)** The log-normal distribution from which we generated synthetic Fc species abundances. **(j)** Comparison of our synthetic distribution with the aggregate distribution of measurements for all datasets and subclass detections (composed of the data shown in panels a-h). In **(j)**, all the log distributions are mean divided before being plotted so that we can directly compare the shapes of the distributions.
(EPS)

**S2 Fig. Binding model maintains antibody Fc prediction performance from synthetic data despite significant errors in the crosslinking constant.** Here, we examined the effect of inaccurate estimates of $K_x^*$ on model prediction accuracy. To do this, we established ground truth initial antibody Fc abundances and $K_x^*$ and then used the antibody-to-signal function to generate synthetic detection signals, as previously described. The key difference in this test is that we provided a deliberately inaccurate $K_x^*$ to the signal-to-antibody function when inferring the antibody Fc abundances. We generated inaccurate $K_x^*$ by multiplying the ground truth $K_x^*$ by a perturbation coefficient. The agreement between the ground truth antibody abundances and the model's inferences from the synthetic data are shown for various perturbation coefficients.
(EPS)

**S3 Fig. A rank of 1 provides the best imputation performance for PCA.** To determine the optimal rank for PCA imputation, 10% of values for a chosen detection were masked in the Zohar et al dataset and PCA was used to impute the missing values[11]. **(a), (c)** Imputation accuracy measured by the **(a)** coefficient of determination and **(c)** Pearson correlation, separated by detection. **(b), (d)** Total Imputation accuracy across all detections measured by **(a)**

coefficient of determination and **(c)** Pearson correlation.
(EPS)

**S4 Fig. Imputation performance is largely invariant to dataset size.** We examine how the imputation performance of the binding model is affected by the total size of the dataset by removing entire rows in the dataset at different ratios, and then passing this reduced dataset to the same imputation pipeline we detailed earlier. At each dataset size, we use the model to impute 50% of the values corresponding to a single column (detection type) and measured the agreement between these inferences and the ground truth. **(a)**, **(b)** Imputation accuracy measured by the **(a)** Pearson correlation and **(b)** coefficient of determination, separated by detection.
(EPS)

**S5 Fig. Direct fucosylation measurements are inconsistent with FcR interaction and effector function measurements. (a)** In Alter *et al [10]*, both detection signals and direct CE IgG fucosylation measurements for IgG targeting one antigen exist. We use our model to infer the IgG fucosylation from the detection signals and then compare these to the CE measurements. **(b)** IgG fucosylation measured by CE versus that inferred by binding model. **(c), (d)** gp120. SF162-targeting IgG fucosylation measured by CE versus detection signal ratios **(c)** FcγRIIIA / α-HIgG1 and **(d)** FcγRIIIA / FcγRIIA. **(e), (f)** gp120.SF162-targeting IgG fucosylation inferred by binding model versus detection signal ratios **(e)** FcγRIIIA / α-HIgG1 and **(f)** FcγRIIIA / FcγRIIA. **(g)** gp120.SF162-targeting IgG detection signal of FcγRIIIA versus gp120.SF162-targeting IgG antibody-dependent natural killer cell activation (ADNKA) measured by MIP1β expression. **(h)** gp120.SF162-targeting IgG detection signal of FcγRIIA versus gp120. SF162-targeting IgG antibody-dependent neutrophil phagocytosis (ADNP). **(i)** CE-measured gp120.SF162-targeting IgG fucosylation versus bisection. The correlation metrics shown are spearman rank correlations, denoted as $r_S$.
(EPS)

## Author Contributions

**Conceptualization:** Aaron S. Meyer.

**Investigation:** Armaan A. Abraham, Zhixin Cyrillus Tan, Priyanka Shrestha, Emily R. Bozich.

**Methodology:** Aaron S. Meyer.

**Software:** Armaan A. Abraham, Zhixin Cyrillus Tan, Aaron S. Meyer.

**Supervision:** Aaron S. Meyer.

**Visualization:** Armaan A. Abraham.

**Writing – original draft:** Armaan A. Abraham.

**Writing – review & editing:** Priyanka Shrestha, Emily R. Bozich, Aaron S. Meyer.

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
