## [Decision Letter · Decision Letter 0]

23 Aug 2024

Dear Prof. Meyer,

Thank you very much for submitting your manuscript "A multivalent binding model infers antibody species from systems serology" for consideration at PLOS Computational Biology.

As with all papers reviewed by the journal, your manuscript was reviewed by members of the editorial board and by several independent reviewers. In light of the reviews (below this email), we would like to invite the resubmission of a significantly-revised version that takes into account the reviewers' comments.

We cannot make any decision about publication until we have seen the revised manuscript and your response to the reviewers' comments. Your revised manuscript is also likely to be sent to reviewers for further evaluation.

Sincerely,

Anders Wallqvist

Academic Editor

PLOS Computational Biology

Rob De Boer

Section Editor

PLOS Computational Biology

Reviewer's Responses to Questions

**Comments to the Authors:**

Reviewer #1: Summary: This work addressed a novel computational model application to provide a more accurate proxy for quantifying antibody abundance and properties. This challenge has been extensively explored in omics research and demonstrated significant potential for application in antibody response profiling, supporting immunological discoveries and vaccine design. By detailing their binding model and validating its implementations with existing antibody profile datasets, the authors highlighted several advantages of their approach, including simplifying serological protocols and advancing the new discovery including the understanding of the relationship between antibody structure and vaccine efficacy.

Major:

1. While inferring true antibody abundance is challenging, the impact of this work could be significantly enhanced by elaborating on existing benchmarks and providing additional performance comparisons beyond the use of raw measurement inputs.

2. The authors utilized synthetic data simulation and incorporated Gaussian-distributed noise to demonstrate their method’s ability to reconstruct the "ground truth" antibody quantity. However, in practical applications, distinguishing between noise and a genuine lack of signal can be challenging. The manuscript would benefit from a discussion on the implications of noise removal for false positive/negative discoveries or from additional simulations that address real-world data complexities.

3. In the introduction, the authors have highlighted the use and development of advanced statistical and machine learning methods in systems serology studies. To enhance the comprehensiveness of this work, it would be beneficial to integrate a discussion on mathematical modeling methods, especially in the context of advancing or comparing these techniques with machine learning applications in immunological profile studies in tasks including signal annotation and missing value imputation.

Minor:

1. Fig 3. c, Fig.7.e-h were labeled with both r and R2. Please explicit what these are in the figure legends or labels for better information delivery.

2. The color scheme in Figure 5 used the same sets of colors for different underlying subgroups; Panel a and c used the same set of x-axis but are ordered differently.

3. An overall color legend for Fig 7. a-d is helpful for visualization.

Reviewer #2: The SARS-CoV-2 vaccine produces a spike antigen to induce an immune response. Abraham et al used computational techniques to find that the efficacy of this vaccine is correlated with the raising of an immune response towards the spike antigen. They also examined HIV. Though it may appear obvious that an immune response must be raised for the vaccine to be efficacious, the authors aim to unveil mechanistic details of this process. The manuscript appears to be an amalgamation of several interesting, but only loosely connected, research tidbits - the readability of future work would be improved with a tighter focus, even if it means a smaller paper. This would also help readers better understand how it applies to their own work, and how they could use it to advance their own work. The admission of shortcomings, and descriptions of small failures (i.e. "At first, we attempted to fit γ, but this made the fitting too flexible, resulting in unrealistic values of 515" and paragraphs such as that one), is a strength of this work, as these bits of knowledge can speed up future research progress for others. Overall, this was an interesting manuscript to read, and the authors did a thorough job with their work. After making some revisions for manuscript clarity, I believe this work should be published in PLoS Computational Biology.

1) The authors mention that they need to estimate the crosslinking constant from a previous estimate. Building a computational model to infer experimental data is tricky to begin with, and building this model based off of estimates of experimental data is moving towards shaky ground. How confident are the authors that this is a reasonable estimate? What can they do to show naive readers that this estimate should be trusted?

2) There are several instances, such as in the abstract and introduction, where the authors imply that they are studying broad mechanistic details of antibody binding and the raising of an immune response, while they actually have a narrow focus of inferring glycosylation patterns of antibodies and deducing which experimental systems serology data points are necessary, and which are unnecessary, in antibody assays. This is valuable in its own right, and the manuscript would be strengthened by the authors narrowing their claims. The current broader focus of the manuscript dilutes and muddies the worthwhile results that the authors have found, and would make it more difficult for an average reader to understand this work. Furthermore, from the results section it appears that these results are only applicable to the SARS-CoV-2 and HIV vaccines, whereas the abstract and introduction imply that the authors used these two vaccines to make a model that can draw broader conclusions about immune responses.

3) "However, both FcγRIIa and FcγRIIb gain a significant bias in their predictions as the number of masked values is increased, as indicated by the increasing R2." Why is this the case? Why do some antibodies gain bias while others don't?

4) The model can infer fucosylation because of the reduction in binding affinity between IgG and FcγRIIIA. Is this fucosylation prediction generalizable to other pairs, such as IgG and FcγRIIA?

5) In one section, the authors mention that "This correlation between HIV infection severity and IgG afucosylation is like the correlation we found between COVID-19 severity and IgG afucosylation." Thus afucosylation can be predictive of COVID-19 severity and HIV infection severity, in those specific cases. However, the authors later state that "We found a high degree of variation in IgG fucosylation by antigen across subjects (Fig. 6d), indicating that antigen-specific IgG fucosylation cannot be inferred using the IgG fucosylation for another antigen from the same pathogen." These two statements appear contradictory. If there is a high degree of variation in IgG fucosylation, then it would appear not to be directly predictive of infection severity. It would be preferable to have a general model that can predict fucosylation regardless of the antigen in question, but is understandable that this is not the case. However, one would expect the conclusions to be the same regardless of the antigen in question (i.e. that afucosylation is predictive of infection severity). If this is not the case, an explanation should be provided for clarity.

6) "This study includes CE IgG fucosylation measurements for the gp120.SF162 HIV antigen10" As written, it appears that the submitted manuscript contains CE IgG fucosylation measurements, and the citation above is a general citation for the gp120.SF162 HIV antigen. This should be rewritten for clarity.

7) The source code is provided on github. However, there is no README or instructions for users to either build the model for themselves, or for users to download prebuilt models. Additionally, it is unclear whether the authors expect users to build models for themselves or download prebuilt models. These should be remedied to improve the usability of the work.

Reviewer #3: In this manuscript, Abraham and colleagues developed a model for inferring Fc structural features from systems serology assays. Having such a model is of great value as it would potential reduce the need for lengthy wet-lab experimentation.

Please see my comments below.

Major

- Title: The term “antibody species” is somewhat unusual. At least to me, it was not clear that it would describe Fc features. Consider rewording.

- Figure 1: it’s not immediately clear from this figure, what the model’s purpose and output is. Consider revising.

- Figure 2: To what extent does the synthetic date recapitulate the key characteristics of experimental data? In other words, how biologically relevant are the simulations?

- Figure 3: what’s the maximum amount of missing data (%) you can infer? What’s the influence of dataset size on imputation capacity?

- Figure 4: Can you clarify whether these measurements are experimentally validated? If not, please add somewhat of a confidence measurement to your predictions (if possible) and maybe weaken the statements attached to this figure a bit.

- Figure 5: Please show the data points in all box plots (for all boxplots in the paper).

- Model: Consider moving some of the mathematical details of model derivation to to the main text. This would help the reader more quickly grasp the main variables of the model.

- Can the authors comment on the batch to batch variability of the model? In other words, are parameters learned from one dataset, applicable to another dataset where the biological sample was the same? Have the authors tried to apply the model to technical and biological replicates to understand whether the parameter inference is robust?

**Have the authors made all data and (if applicable) computational code underlying the findings in their manuscript fully available?**

Reviewer #1: Yes

Reviewer #2: **No: **It it not clear whether their model(s) are provided on Github, or where they are located if so. Instructions for using the model(s) are not provided.

Reviewer #3: None

PLOS authors have the option to publish the peer review history of their article (what does this mean?). If published, this will include your full peer review and any attached files.

Reviewer #1: No

Reviewer #2: **Yes: **Christian G. Seitz

Reviewer #3: No
---

## [Decision Letter · Decision Letter 1]

20 Nov 2024

Dear Prof. Meyer,

We are pleased to inform you that your manuscript 'A multivalent binding model infers antibody Fc species from systems serology' has been provisionally accepted for publication in PLOS Computational Biology.

Best regards,

Anders Wallqvist

Academic Editor

PLOS Computational Biology

Rob De Boer

Section Editor

PLOS Computational Biology

Feilim Mac Gabhann

Editor-in-Chief

PLOS Computational Biology

Jason Papin

Editor-in-Chief

PLOS Computational Biology

Reviewer's Responses to Questions

**Comments to the Authors:**

Reviewer #1: The authors have adequately addressed my prior comments and I believe the work is ready for publication.

Reviewer #2: This reviewer would like to thank Abraham et al for the time they spent carefully addressing each requested revision. It is the strong viewpoint of this reviewer that the revisions have strengthened the manuscript, clarifying important points and expanding on difficult concepts when necessary. It is especially important that the code is now available on github, with instructions, for the community to use. Job well done.

Reviewer #3: The authors have addressed all of my comments.

**Have the authors made all data and (if applicable) computational code underlying the findings in their manuscript fully available?**

Reviewer #1: Yes

Reviewer #2: Yes

Reviewer #3: Yes

PLOS authors have the option to publish the peer review history of their article (what does this mean?). If published, this will include your full peer review and any attached files.

Reviewer #1: No

Reviewer #2: No

Reviewer #3: No

---

## [Editor Report · Acceptance letter]

15 Dec 2024

PCOMPBIOL-D-24-01154R1 

A multivalent binding model infers antibody Fc species from systems serology

Dear Dr Meyer,

I am pleased to inform you that your manuscript has been formally accepted for publication in PLOS Computational Biology. Your manuscript is now with our production department and you will be notified of the publication date in due course.

With kind regards,

Zsofia Freund
